# Bridging the Gap: Unifying the Training and Evaluation of Neural Network Binary Classifiers

**Nathan Tsoi**
Yale University
nathan.tsoi@yale.edu

**Kate Candon**
Yale University
kate.candon@yale.edu

**Deyuan Li**
Yale University
deyuan.li@yale.edu

**Yofti Milkessa**
Yale University
yofti.milkessa@yale.edu

**Marynel Vázquez**
Yale University
marynel.vazquez@yale.edu

## Abstract

While neural network binary classifiers are often evaluated on metrics such as Accuracy and $F_1$-Score, they are commonly trained with a cross-entropy objective. How can this training-evaluation gap be addressed? While specific techniques have been adopted to optimize certain confusion matrix based metrics, it is challenging or impossible in some cases to generalize the techniques to other metrics. Adversarial learning approaches have also been proposed to optimize networks via confusion matrix based metrics, but they tend to be much slower than common training methods. In this work, we propose a unifying approach to training neural network binary classifiers that combines a differentiable approximation of the Heaviside function with a probabilistic view of the typical confusion matrix values using soft sets. Our theoretical analysis shows the benefit of using our method to optimize for a given evaluation metric, such as $F_1$-Score, with soft sets. Also, our extensive experiments show the effectiveness of our approach in several domains.

## 1 Introduction

Neural network binary classifiers output a probability $p \in [0, 1]$ which is often used at training time to optimize model parameters using the binary cross-entropy (BCE) loss. The network's output $p$ can also be translated to a binary value $\{0, 1\}$ indicating set membership to the negative or positive class. To determine set membership, the Heaviside step function $H$ is commonly used with a threshold $\tau$, where $p \geq \tau$ are considered positive classification outcomes. This notion of set membership is often used in evaluation metrics for binary classifiers.

It is a common assumption that optimizing a network via the desired evaluation metric is preferable to optimizing a surrogate objective [12, 18, 29, 33]. Unfortunately, optimizing a network using the desired evaluation metric, such as $F_1$-Score, is typically not feasible. The Heaviside step function used to compute confusion matrix set membership in terms of true positives, false negatives, false positives, and true negatives, has a gradient with properties not conducive to optimization via gradient descent. The Heaviside function's gradient is not defined at the threshold $\tau$ and is zero everywhere else.

Empirical results (e.g., [12]) show that the training-evaluation gap hinders evaluation performance. The gap means that in many applications, the metric differs significantly from the surrogate loss [33] and there may not be a strong correlation between minimizing a surrogate loss and improving an evaluation metric [29]. Also, it could mean that classifiers are solving "the wrong problem" when optimizing a surrogate loss, leading to sub-optimal evaluation performance [18].

36th Conference on Neural Information Processing Systems (NeurIPS 2022).

To address the gap between training and evaluating neural network binary classifiers, we propose a method to make confusion-matrix based evaluation metrics usable for backpropagation. Specifically, we propose the use of a differentiable approximation of the Heaviside step function along with confusion matrix values computed using the notion of soft set membership. A desired evaluation metric can then be made differentiable by calculating it over the soft-set confusion matrix, rather than using the traditional confusion matrix values.

Our main contributions are: 1) a novel method for training neural network binary classifiers that allows for the optimization of confusion-matrix based evaluation metrics with soft sets (Sec. 3); 2) a theoretical analysis of our method (Sec. 4); and 3) the application of our approach to various domains with varying levels of class imbalance, showing its flexibility and superior performance compared to several baseline methods (Sec. 5). We provide an open-source implementation of our method for reproducibility.[1]

## 2   Preliminaries

In binary classification via neural networks, a step function is required to transform the network's output to a binary value. A common choice is the Heaviside step function with a threshold value $\tau$:

$$H(p, \tau) = \begin{cases} 1 & p \geq \tau \\ 0 & p < \tau \end{cases} \tag{1}$$

Confusion matrix set membership is then computed for a prediction $p$ and ground truth label $y$ via:

$$
tp(p, y, \tau) = \begin{cases} H(p, \tau) & y = 1 \\ 0 & \text{otherwise} \end{cases} \qquad
fn(p, y, \tau) = \begin{cases} 1 - H(p, \tau) & y = 1 \\ 0 & \text{otherwise} \end{cases}
$$

$$
fp(p, y, \tau) = \begin{cases} H(p, \tau) & y = 0 \\ 0 & \text{otherwise} \end{cases} \qquad
tn(p, y, \tau) = \begin{cases} 1 - H(p, \tau) & y = 0 \\ 0 & \text{otherwise} \end{cases}
\tag{2}
$$

Consider a set of predictions $p \in [0, 1]$, ground truth labels $y \in \{0, 1\}$ and threshold value $\tau \in (0, 1)$, e.g., $\{(p_1, y_1, \tau), (p_2, y_2, \tau), ..., (p_n, y_n, \tau)\}$. For $n$ samples, the cardinality of each confusion matrix set is then computed as:

$$|TP| = \sum_{i=1}^{n} tp(p_i, y_i, \tau) \quad |FN| = \sum_{i=1}^{n} fn(p_i, y_i, \tau) \quad |FP| = \sum_{i=1}^{n} fp(p_i, y_i, \tau) \quad |TN| = \sum_{i=1}^{n} tn(p_i, y_i, \tau)$$

Common classification metrics are based on these cardinalities. For example, Precision $= |TP|/(|TP| + |FP|)$ is the proportion of positive predictions that are true positive results. Recall $= |TP|/(|TP| + |FN|)$ indicates the proportion of positive examples that are correctly identified. These two metrics represent a trade-off between classifier objectives; it is generally undesirable to optimize or evaluate for one while ignoring the other [17]. This makes summary evaluation metrics that balance between these trade-offs popular. Commonly used metrics include:

$$\text{Accuracy} = \frac{|TP| + |TN|}{|TP| + |TN| + |FP| + |FN|} \qquad F_1\text{-Score} = \frac{2}{\text{precision}^{-1} + \text{recall}^{-1}} \tag{3}$$

Accuracy is the rate of correct predictions to all predictions. $F_1$-Score is a specific instance of $F_\beta$-Score, which is the weighted harmonic mean of precision and recall: $F_\beta$-Score $= (1 + \beta^2) \cdot (\text{precision} \cdot \text{recall})/(\beta^2 \cdot \text{precision} + \text{recall})$. The value $\beta$ indicates that recall is considered $\beta$ times more important than precision.

Some metrics like $F_\beta$-Score are usually computed at a specific threshold $\tau$ whereas others are computed over a range of $\tau$ values. For example, the area under the receiver operating characteristic curve (AUROC) is a commonly used ranking performance measure defined in terms of the true positive rate (TPR) and false positive rate (FPR), each a function of $\tau$: $\int_{\infty}^{-\infty} \text{TPR}(\tau)\text{FPR}'(\tau) \, d\tau$.

While metrics that rely on confusion matrix set values are commonly used for evaluation, as shown in Fig. 1 (right), it is difficult to use them as a loss during training. These metrics rely on the Heaviside

---

[1]Source code: `https://github.com/nathantsoi/btg`.

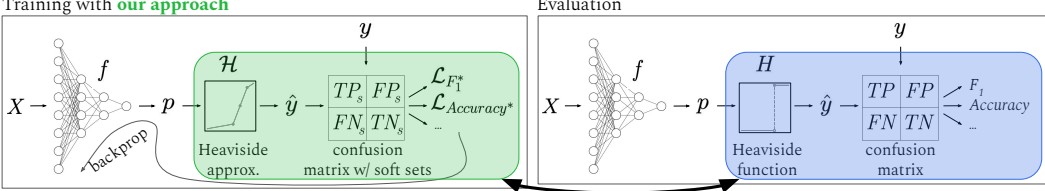

Figure 1: The proposed method. Binary classifiers are typically trained with the BCE loss and then evaluated on a confusion-matrix based metric. We propose to bridge this gap between training and evaluation by optimizing model parameters based on a metric that is computed over a soft-set confusion matrix, which is differentiable. This is done using a differentiable approximation of the Heaviside step function.

step function, the derivative of which is undefined at the threshold $\tau$ and zero everywhere else. This means that these metrics do not have a derivative useful to backpropagate errors through the network.

When a binary classification neural network is trained on a loss function that is different from the evaluation metric, such as BCE, the network parameters are unlikely to be optimal for the desired evaluation metric. As shown in our theoretical analysis and experimental results, when the goal is to balance various confusion matrix set values, e.g., with $F_1$-Score, performance is improved by using our proposed method to optimize $F_1$-Score.

## 3    Method

We propose a method (Fig. 1) that aims to better unify the training and evaluation steps of binary neural network classifiers whose performance is measured with metrics based on the confusion matrix. Our method has two main steps. First, the Heaviside step function, $H$, is approximated with a function $\mathcal{H}$ useful for optimization via gradient descent. Then, $\mathcal{H}$ is used to compute a soft version of set membership in the confusion matrix. With the soft-set confusion matrix values, we can then compute desired confusion-matrix based metrics, such as $F_1$-score, using their standard formula. This approach makes the metrics end-to-end differentiable so that they can be used as a training loss.

### 3.1    Heaviside Approximation

A useful Heaviside approximation, $\mathcal{H}$, has a non-zero gradient: $\mathcal{H}'(p, \tau) \neq 0$ , $\forall \tau$. Also, it ensures that for a single example, the classifier predicts positive and negative probabilities which sum to 1. Like $H$, the approximation should meet the properties of a cumulative distribution function (CDF) which ensures it is right-continuous, non-decreasing, with outputs in $[0, 1]$ following:

$$\lim_{p \to 0} \mathcal{H}(p, \tau) = 0 \quad \forall \tau \qquad\qquad \lim_{p \to 1} \mathcal{H}(p, \tau) = 1 \quad \forall \tau \qquad\qquad (4)$$

**Sigmoid approximation:** One approximation method for $H$, proposed by [23], is the sigmoid function $s_0(k; p) = (1 + e^{-kp})^{-1}$. We reparameterize $s_0$ to account for $\tau$, so that it can be used for varying thresholds: $\mathcal{H}^s(k, p, \tau) = (1 + e^{-k(p-\tau)})^{-1}$. A challenge with the sigmoid approximation is that as $k$ increases, the sigmoid function better approximates the Heaviside function, but the derivative is close to zero over a larger range of valid inputs. Another challenge is that when $\tau$ is close to 0 or 1, $\mathcal{H}^s$ does not approach $H$ as the input approaches 0 or 1. These limitations as well as the hyperparameters $k$ and $\tau$ are further discussed in the Supplementary Material (Sec. 2).

**Linear approximation:**    An alternative approximation of $H$ is a five-point linearly interpolated function (Fig. 1, left).    This approximation $\mathcal{H}^l$, is defined over $[0, 1]$ and is parameterized by a given threshold $\tau$ and a slope parameter $\delta$, which define three linear segments with slopes $m_1$, $m_2$, and $m_3$.    The slope of each line segment is:

$$m_1 = \frac{\delta}{\tau - \frac{\tau_m}{2}} \qquad\qquad m_2 = \frac{1 - 2\delta}{\tau_m} \qquad\qquad m_3 = \frac{\delta}{1 - \tau - \frac{\tau_m}{2}}$$

with $\tau_m = \min\{\tau, 1 - \tau\}$ in order to ensure a gradient suitable for backpropagation. The linear

Heaviside approximation is thus given by:

$$\mathcal{H}^l = \begin{cases} p \cdot m_1 & \text{if } p < \tau - \frac{\tau_m}{2} \\ p \cdot m_3 + (1 - \delta - m_3(\tau + \frac{\tau_m}{2})) & \text{if } p > \tau + \frac{\tau_m}{2} \\ p \cdot m_2 + (0.5 - m_2\tau) & \text{otherwise} \end{cases} \tag{5}$$

Considering the threshold $\tau$ in the formulation of $\mathcal{H}^l$ ensures $\mathcal{H}^l(p = \tau, \tau) = 0.5$ while maintaining the limits in Eq. (4). See the Supplementary Material, Sec. 2.2, for the derivation of Eq. (5).

Recently, a similar linear function was implemented as an activation function to enforce meaningful logical outputs in logical neural networks [30]. However, rather than using the linear function for activation within the larger model architecture, we use it in the training objective of a classifier.

## 3.2 Soft Sets

We use soft sets, a generalization of fuzzy sets [25], to compose differentiable versions of confusion-matrix based evaluation metrics useful for backpropagation. A soft set $\mu$ in $U$, the initial universal set, is defined by the membership function $\mu : U \to [0, 1]$. For $x \in U$, the set membership function $\mu(x)$ specifies a degree of belonging for $x$ to set $u$.

We use the notion of soft sets to compute confusion-matrix based metrics with a Heaviside approximation in place of the typically used strict sets. Soft set membership corresponds to the degree to which a sample tuple $(p, y, \tau)$ belongs to a confusion matrix set. We define the soft confusion matrix set membership functions for prediction and ground truth examples $(p, y)$ relative to $\tau$ as:

$$tp_s(p, y, \tau) = \begin{cases} \mathcal{H}(p, \tau) & y = 1 \\ 0 & \text{otherwise} \end{cases} \qquad fn_s(p, y, \tau) = \begin{cases} 1 - \mathcal{H}(p, \tau) & y = 1 \\ 0 & \text{otherwise} \end{cases}$$

$$fp_s(p, y, \tau) = \begin{cases} \mathcal{H}(p, \tau) & y = 0 \\ 0 & \text{otherwise} \end{cases} \qquad tn_s(p, y, \tau) = \begin{cases} 1 - \mathcal{H}(p, \tau) & y = 0 \\ 0 & \text{otherwise} \end{cases} \tag{6}$$

Whereas the values of the confusion matrix $(tp, fn, fp, tn)$ are the sum of zeros and ones, and the values of the soft confusion matrix $(tp_s, fn_s, fp_s, tn_s)$ are the sum of continuous values in $[0, 1]$, rather than integers.

After computing per-sample soft set values, specific metrics can be approximated by summing over the relevant elements of the confusion matrix. For instance, we approximate precision as $|TP_s|/(|TP_s| + |FP_s|)$, where $|TP_s| = \sum_{i=1}^{n} tp_s(p_i, y_i, \tau)$ and $|FP_s| = \sum_{i=1}^{n} fp_s(p_i, y_i, \tau)$ for $n$ samples. In practice, this summation occurs at training time over mini-batches while optimizing via gradient descent. Because gradient descent and its variants expect a small but representative sample of the broader data [5], the proposed method also expects a representative sample.

**Decomposability:** Metrics based on confusion matrix set values are considered non-decomposable. Non-decomposable metrics cannot be calculated per datapoint and are not additive across subsets of data [19]. We acknowledge that our optimization method of mini-batch stochastic gradient descent (SGD) does not provide an unbiased estimator on non-decomposable metrics. However, this is common in neural network training [16]. In practice, we find that large enough mini-batches provide a representative sample for confusion matrix based metrics, allowing the use of mini-batch SGD in our approach. Moreover, our method does not limit maximum batch size for training, unlike the adversarial approach to optimize $F_1$-Score proposed by [14].

**Metrics and losses:** Any metric composed of confusion matrix set values ($TP$, $FP$, $FN$, $TN$) can be approximated using our proposed method and used as a training loss. In our experiments, we train and evaluate on Accuracy, $F_1$-Score, AUROC, and $F_\beta$-Score [31]. We chose these metrics to show the flexibility of our approach. Accuracy, $F_1$-Score and $F_\beta$-Score may be selected based on class imbalance and resulting tradeoffs. AUROC illustrates that our method can be used when computation over a range of thresholds, $\tau$, is required.

## 4 Theoretical Grounding

In this section, we first show the Lipschitz continuity of a variety of soft-set based metrics under the proposed Heaviside function approximations. This indicates that when such metrics are used as the

loss in a neural network, the difference between successive losses is bounded across iterations of stochastic gradient descent. We then show, under certain assumptions, that metrics computed over the soft-set confusion matrix values are asymptotically similar to the true metric.

## 4.1 Lipschitz Continuity of Metrics Based on Soft-Set Confusion Matrix Values

A function $f$ is considered Lipschitz continuous if there exists some constant $K$ such that for all $x_1, x_2$ in the domain, $|f(x_1) - f(x_2)| \leq K|x_1 - x_2|$. Lipschitz continuous functions are always themselves continuous.

**Theorem 4.1.** *The linear Heaviside approximation $\mathcal{H}^l$ is Lipschitz continuous with Lipschitz constant* $M = \max\{m_1, m_2, m_3\}$.

$\mathcal{H}^l$ is continuous because each piecewise linear component is continuous and $\mathcal{H}^l$ is defined to be continuous at the two points $p = \tau \pm \frac{\tau_m}{2}$. Also, by construction, the slope of any secant line is positive and bounded by $M = \max\{m_1, m_2, m_3\}$. Thus, $\mathcal{H}^l$ is Lipschitz continuous with Lipschitz constant $M$. Please see the full proof in Sec. 1.1 of the Supplementary Material. Note that the $\mathcal{H}^s$ is also Lipschitz continuous [32].

**Theorem 4.2.** *Every entry of the soft-sets confusion matrix based on the Heaviside approximations is Lipschitz continuous in the output of a neural network.*

The proof of Theorem 4.2 is in the Supplementary Material (Sec. 1.1).

The value of a Lipschitz continuous loss function composed of soft-set confusion matrix values is Lipschitz continuous. This is because compositions of Lipschitz continuous functions are also Lipschitz continuous. Following [9], we note that the confusion-matrix based metrics Accuracy, Balanced Accuracy, $F_\beta$, Jaccard, and G-Mean are all Lipschitz continuous. These metrics are also Lipschitz continuous under our proposed method computed via soft sets.

Lipschitz continuity of the loss function in a neural network optimized via stochastic gradient descent indicates convergence without extreme variations in losses throughout training. Let $K$ be the Lipschitz constant for the objective function $\ell(w)$ on network weights $w$. Because of Lipschitz continuity, when updating $w_{i+1} \to w_i - \alpha_i \ell'(w_i)$ using stochastic gradient descent with a learning rate $\alpha_i$, a small local change in the weights $|w_{i+1} - w_i| = \alpha_i |\ell'(w_i)|$ corresponds to a small local change in the value of the objective function of $|\ell(w_{i+1}) - \ell(w_i)| \leq K\alpha_i |\ell'(w_i)|$.

## 4.2 Approximation of Confusion-Matrix Based Metrics with Soft Sets

We provide a statistical analysis showing that in the limit, as the number of examples goes to infinity, $F_1$-Score calculated with soft sets approximates the expected true $F_1$-score under a set of assumptions. Similar proofs for Accuracy and AUROC are provided in the Supplementary Material (Sec. 1.2).

Consider a dataset of size $n$ with $\{x_1, ..., x_n\}$ examples and $\{y_1, ..., y_n\}$ labels. Suppose a network outputs a probability $p_i$ that the label $y_i = 1$. In this section, since the specific outputs $p_i$ are unknown and may change across iterations, we assume $p_i$ is a random variable. When calculating $F_1$-Score, $p_i$ is passed through the Heaviside function $H$, which generates an output $\hat{y}_i^H = H(p_i, \tau)$, where $\hat{y}_i^H \in \{0, 1\}$. The $F_1$-Score can be expressed as:

$$F_1 = \frac{\sum_{i=1}^n y_i \hat{y}_i^H}{\sum_{i=1}^n y_i \hat{y}_i^H + \frac{1}{2} \sum_{i=1}^n ((1 - y_i)\hat{y}_i^H + y_i(1 - \hat{y}_i^H))} = \frac{2 \sum_{i=1}^n y_i \hat{y}_i^H}{\sum_{i=1}^n (y_i + \hat{y}_i^H)} \tag{7}$$

To calculate $F_1$-Score with soft sets ($F_1^s$), $p_i$ is passed through the Heaviside approximation $\mathcal{H}$, which generates an output $\hat{y}_i^{\mathcal{H}} = \mathcal{H}(p_i, \tau)$, where $\hat{y}_i^{\mathcal{H}} \in [0, 1]$. $F_1$-Score with soft-sets can be expressed as:

$$F_1^s = \frac{2 \sum_{i=1}^n y_i \hat{y}_i^{\mathcal{H}}}{\sum_{i=1}^n (y_i + \hat{y}_i^{\mathcal{H}})} \tag{8}$$

Consider a network trained on a dataset with $rn$ positive and $(1 - r)n$ negative elements, where $r \in [0, 1]$ is some constant. Suppose this classifier correctly classifies any positive example as a true positive with probability $u$ and any negative example as a false positive with probability $v$. Also, assume that all classifications are independent. Because $F_1$-Score is calculated with discrete $\hat{y}_i^H$, we

assume that the classifier will classify examples as a random variable $\hat{y}_i^H \sim \text{Bernoulli}(uy_i + v(1 - y_i))$. Thus, $\hat{y}_i^H \sim \text{Bernoulli}(u)$ if $y_i = 1$, and $\hat{y}_i^H \sim \text{Bernoulli}(v)$ if $y_i = 0$.

Since $F_1^s$ can take on continuous values in $[0, 1]$, we consider that $\hat{y}_i^{\mathcal{H}}$ is a random variable drawn from a Beta distribution, which has support $[0, 1]$. In particular, assume $\hat{y}_i^{\mathcal{H}} \sim \text{Beta}(\alpha_u y_i + \alpha_v (1 - y_i), \beta_u y_i + \beta_v(1 - y_i))$. Hence, $\hat{y}_i^{\mathcal{H}} \sim \text{Beta}(\alpha_u, \beta_u)$ if $y_i = 1$, and $\hat{y}_i^{\mathcal{H}} \sim \text{Beta}(\alpha_v, \beta_v)$ if $y_i = 0$. Let $\frac{\alpha_u}{\alpha_u + \beta_u} = u$ and $\frac{\alpha_v}{\alpha_v + \beta_v} = v$, so for any $i$, $\mathbb{E}\left[\hat{y}_i^{\mathcal{H}}\right] = u$ if $y_i = 1$, and $\mathbb{E}\left[\hat{y}_i^{\mathcal{H}}\right] = v$ if $y_i = 0$.

Under the above assumptions, both $F_1$ and $F_1^s$ have the same average classification correctness: for any given $i$, $\mathbb{E}\left[\hat{y}_i^H\right] = \mathbb{E}\left[\hat{y}_i^{\mathcal{H}}\right]$. Also, there exists some $\alpha_u, \beta_u, \alpha_v, \beta_v$ such that the distributions of $\hat{y}_i^H = H(p_i, \tau)$ and $\hat{y}_i^{\mathcal{H}} = \mathcal{H}(p_i, \tau)$ can both hold simultaneously under the same network for all $i$.

If we let $U \sim \text{Binomial}(nr, u)$ and $V \sim \text{Binomial}(n(1-r), v)$ be the independent random variables denoting the number of true positives and false positives in a sequence of $n$ independent predictions, then $F_1$ from Eq. (7) becomes:

$$F_1 = \frac{2 \sum_{i=1}^n y_i \hat{y}_i^H}{\sum_{i=1}^n (y_i + \hat{y}_i^H)} = 2 \left( \frac{\sum_{i=1}^n y_i \hat{y}_i^H}{nr + \sum_{i=1}^n \hat{y}_i^H} \right) = 2 \left( \frac{U}{nr + U + V} \right) = 2 \left( \frac{U/n}{r + U/n + V/n} \right) \quad (9)$$

By the Strong Law of Large Numbers, $\frac{1}{nr} U \xrightarrow{\text{a.s.}} u$ and $\frac{1}{n(1-r)} V \xrightarrow{\text{a.s.}} v$ both converge with probability 1 as $n \to \infty$. Hence, $\frac{U}{n} \xrightarrow{\text{a.s.}} ru$ and $\frac{V}{n} \xrightarrow{\text{a.s.}} (1 - r)v$. We therefore have, from the Continuous Mapping Theorem, that as $n \to \infty$:

$$F_1 = 2 \left( \frac{U/n}{r + U/n + V/n} \right) \xrightarrow{\text{a.s.}} 2 \left( \frac{ru}{r + ru + (1 - r)v} \right) = \frac{2ru}{r + ru + v - rv} \quad (10)$$

For $F_1^s$, let $U^s = \sum_{y_i=1} \hat{y}_i^{\mathcal{H}}$ and $V^s = \sum_{y_i=0} \hat{y}_i^{\mathcal{H}}$ be the independent random variables denoting the total amount of true positives and false positives in the soft set case. Then, $F_1^s$ from Eq. (8) becomes:

$$F_1^s = \frac{2 \sum_{i=1}^n y_i \hat{y}_i^{\mathcal{H}}}{\sum_{i=1}^n (y_i + \hat{y}_i^{\mathcal{H}})} = 2 \left( \frac{\sum_{i=1}^n y_i \hat{y}_i^{\mathcal{H}}}{nr + \sum_{i=1}^n \hat{y}_i^{\mathcal{H}}} \right) = 2 \left( \frac{U^s}{nr + U^s + V^s} \right) = 2 \left( \frac{U^s/n}{r + U^s/n + V^s/n} \right) \quad (11)$$

Since $U^s$ is the sum of $nr$ i.i.d. random variables distributed as $\text{Beta}(\alpha_u, \beta_u)$, which has mean $\frac{\alpha_u}{\alpha_u + \beta_u}$, by the Strong Law of Large Numbers, $\frac{1}{nr} U^s \xrightarrow{\text{a.s.}} \frac{\alpha_u}{\alpha_u + \beta_u} = u$. Similarly, $\frac{1}{n(1-r)} V^s \xrightarrow{\text{a.s.}} \frac{\alpha_v}{\alpha_v + \beta_v} = v$ also converges with probability 1 as $n \to \infty$. Hence, $\frac{U^s}{n} \xrightarrow{\text{a.s.}} ru$ and $\frac{V^s}{n} \xrightarrow{\text{a.s.}} (1 - r)v$. We therefore have, from the Continuous Mapping Theorem, that as $n \to \infty$,

$$F_1^s = 2 \left( \frac{U^s/n}{r + U^s/n + V^s/n} \right) \xrightarrow{\text{a.s.}} 2 \left( \frac{ru}{r + ru + (1 - r)v} \right) = \frac{2ru}{r + ru + v - rv} \quad (12)$$

Thus, $F_1$ and $F_1^s$ both converge almost surely to the same value as $n \to \infty$. Since $0 \le \frac{2ru}{r+ru+v-rv} \le 1$ is bounded, $\mathbb{E}[F_1], \mathbb{E}[F_1^s] \to \frac{2ru}{r+ru+v-rv}$ by the Bounded Convergence Theorem. This means that the $F_1^s$ value is an asymptotically unbiased estimator for the expected true $F_1$-Score, and we expect average $F_1$-Score values to converge to $F_1^s$ as $n \to \infty$, under our setup. However, the $F_1$-Score computed from soft sets is not an unbiased estimator for the expected true $F_1$-Score for finite $n$.

The proof of almost sure convergence generalizes for any metric that is a continuous function in the ratio of each entry of the confusion matrix to $n$, as shown in the Supplementary Material (Sec. 1.2). This suggests that optimizing over the desired evaluation metric using our proposed method (e.g., $F_1$-Score computed with soft sets) is justified, as the loss should follow our true final loss closely for a large enough $n$. However, note that some metrics may be poor indicators of true classifier performance, such as Accuracy when data is imbalanced [17]. In such cases, it may not be desirable to optimize for such metrics using our method, as further discussed in the next section.

# 5   Experiments

This section presents three experiments to (1) evaluate the performance of our approach against several baselines on tabular data, (2) evaluate our approach on higher-dimensional image data, and (3) evaluate the ability of our method to balance precision and recall during training.

**Datasets:** Experiments were conducted on five publicly available datasets in a variety of domains and were chosen for their varying levels of class imbalance as explained later. All datasets were minimally pre-processed and split into separate train, test, and validation sets. See Sec. 4.1 of the Supplementary Material for details.

**Architecture and training:** Our experiments aim to fairly evaluate our method using different confusion-matrix based objective functions. Therefore, the same network architecture and training scheme was used unless otherwise noted. Performance was evaluated over 10 repeated trials to control for the effects of random weight initialization. We report the mean of results evaluated over threshold values $T = \{0.1, 0.2, ..., 0.9\}$ for metrics that require a threshold choice at evaluation. The details of the network architecture and training scheme are in Sec. 4.2 of the Supplementary Material.

**Baselines:** We trained networks using the typical BCE loss as well as two existing approaches for optimizing specific confusion matrix metrics: an adversarial approach for $F_1$-score [14], and using an approximation of the Wilcoxon-Mann-Whitney (WMW) statistic for AUROC [39]. Also, the Supplementary Material provides comparisons with other, less related methods for binary classification.

## 5.1   Experiments on Tabular Data

We evaluate our proposed approach using four tabular datasets with different levels of class imbalance. The CocktailParty dataset [40] has a 30.29% positive class balance making it the most class-balanced dataset of those considered in this experiment. Salary classification data in the Adult dataset, from the UCI Machine Learning Repository [11], has a 23.93% positive class balance. Classifications of microcalcifications in the Mammography dataset [38] is heavily skewed with a 2.32% positive class balance. Lastly, the Kaggle Credit Card Fraud Detection dataset [36] has the most extreme class balance with only a 0.17% positive class balance.

We compare baseline methods against the performance of neural networks trained using our method to optimize $F_1$-Score, Accuracy, and AUROC. We instantiated our method using the Heaviside approximations and soft sets. Results using the linear approximation are presented in Table 1. See the Supplementary Material, Sec. 5, for results using the sigmoid approximation, which were comparable.

$F_1$ *computed with soft sets.* Our method of optimizing $F_1$ over the soft-set confusion matrix outperforms baselines when evaluated on $F_1$-Score for all datasets. In Table 1, line (1) has a higher $F_1$-Score than lines (2) and (6). Additionally, even when evaluated on the other metrics (Accuracy and AUROC), networks trained using our method on $F_1$ perform similarly or better than BCE.

*Accuracy computed with soft sets.* Our method of optimizing Accuracy over the soft-set confusion matrix has comparable performance to the BCE baseline when evaluated on Accuracy for all datasets. In Table 1, line (3) has comparable Accuracy to line (6). Note that Accuracy can reward prediction of only the dominant-class in imbalanced datasets [17], potentially resulting in no positive predictions and an $F_1$-Score of 0, as in line (3) of Table 1 for the Mammography dataset.

Our theoretical analysis (Sec. 4.2) indicates that our proposed method of optimizing confusion-matrix based metrics using soft-set $F_1$ or soft-set Accuracy should converge to the evaluation metric ($F_1$ or Accuracy respectively) in the limit. The alternative of using BCE loss performs comparably to our proposed method in some cases; however, $\mathrm{BCE} = -\frac{1}{n} \sum_{i=1}^{n} (y_i \log p_i + (1 - y_i) \log(1 - p_i))$ and Accuracy $= (|TP| + |TN|)/n$ are different expressions that don't generally converge to each other as $n \to \infty$.

*AUROC computed with soft sets.* Neither optimizing AUROC using our method nor optimizing AUROC via the WMW statistic [39] consistently outperforms traditional BCE. In Table 1, lines (4) and (5) have lower $F_1$-Score, Accuracy, and AUROC than line (6). This may be due in part to the AUROC metric's challenge with scale assumptions [15]. Moreover, AUROC is not linear-fractional [21], unlike $F_\beta$-Score and Accuracy. Therefore, AUROC does not necessarily have an unbiased estimator of gradient direction [1], which may also contribute to lackluster performance.

Table 1: Losses (rows): $F_1$, Accuracy, and AUROC via the proposed method (*) using the linear approximation; $F_1$-Score† via adversarial approach [14] and AUROC‡ via WMW statistic [39]. Bold indicates performance better than or equal to the BCE baseline.

| | | CocktailParty ($\mu \pm \sigma$) | | | Adult ($\mu \pm \sigma$) | | |
| | Loss | $F_1$-Score | Accuracy | AUROC | $F_1$-Score | Accuracy | AUROC |
|---|---|---|---|---|---|---|---|
| (1) | $F_1$* | **0.75 ± 0.01** | **0.85 ± 0.01** | **0.82 ± 0.01** | **0.63 ± 0.02** | 0.78 ± 0.04 | **0.78 ± 0.02** |
| (2) | $F_1$† | 0.30 ± 0.06 | 0.76 ± 0.01 | 0.60 ± 0.02 | 0.16 ± 0.02 | 0.78 ± 0.00 | 0.55 ± 0.01 |
| (3) | Accuracy* | **0.70 ± 0.02** | **0.85 ± 0.01** | **0.78 ± 0.01** | 0.35 ± 0.04 | **0.81 ± 0.01** | 0.61 ± 0.02 |
| (4) | AUROC* | 0.51 ± 0.01 | 0.41 ± 0.01 | 0.57 ± 0.00 | **0.42 ± 0.01** | 0.32 ± 0.02 | 0.55 ± 0.01 |
| (5) | AUROC‡ | 0.01 ± 0.03 | 0.70 ± 0.03 | 0.50 ± 0.00 | 0.00 ± 0.00 | 0.76 ± 0.00 | 0.50 ± 0.00 |
| (6) | BCE | 0.70 ± 0.02 | 0.85 ± 0.01 | 0.78 ± 0.01 | 0.26 ± 0.06 | 0.80 ± 0.01 | 0.58 ± 0.02 |

| | | Mammography ($\mu \pm \sigma$) | | | Kaggle ($\mu \pm \sigma$) | | |
| | Loss | $F_1$-Score | Accuracy | AUROC | $F_1$-Score | Accuracy | AUROC |
|---|---|---|---|---|---|---|---|
| (1) | $F_1$* | **0.63 ± 0.04** | 0.98 ± 0.00 | **0.78 ± 0.03** | **0.83 ± 0.02** | **1.00 ± 0.00** | **0.90 ± 0.02** |
| (2) | $F_1$† | 0.46 ± 0.08 | 0.98 ± 0.00 | 0.66 ± 0.04 | **0.76 ± 0.06** | **1.00 ± 0.00** | **0.83 ± 0.04** |
| (3) | Accuracy* | 0.00 ± 0.00 | 0.97 ± 0.00 | 0.50 ± 0.00 | **0.62 ± 0.33** | **1.00 ± 0.00** | **0.78 ± 0.15** |
| (4) | AUROC* | 0.11 ± 0.01 | 0.18 ± 0.04 | 0.57 ± 0.02 | 0.06 ± 0.01 | 0.11 ± 0.00 | 0.55 ± 0.00 |
| (5) | AUROC‡ | 0.00 ± 0.01 | 0.88 ± 0.12 | 0.50 ± 0.00 | 0.00 ± 0.00 | 0.93 ± 0.15 | 0.50 ± 0.00 |
| (6) | BCE | 0.56 ± 0.11 | 0.99 ± 0.00 | 0.71 ± 0.06 | 0.50 ± 0.33 | 1.00 ± 0.00 | 0.73 ± 0.16 |

Table 2: Losses (rows): $F_1$ and Accuracy via our proposed method. Bold indicates performance better than or equal to BCE baseline.

| | | CIFAR-10-Transportation ($\mu \pm \sigma$) | | CIFAR-10-Frog ($\mu \pm \sigma$) | |
| | Loss | $F_1$-Score | Accuracy | $F_1$-Score | Accuracy |
|---|---|---|---|---|---|
| (1) | $F_1$* | **0.91 ± 0.00** | **0.93 ± 0.00** | **0.73 ± 0.01** | **0.95 ± 0.00** |
| (2) | Accuracy* | **0.92 ± 0.00** | **0.93 ± 0.00** | **0.74 ± 0.01** | **0.95 ± 0.00** |
| (3) | BCE | 0.88 ± 0.01 | 0.91 ± 0.00 | 0.59 ± 0.04 | 0.94 ± 0.00 |

Note that in comparison to [14], the adversarial approach $F_1$† did not perform as well in our experiments. The difference could be due to different datasets, and because we used a PyTorch implementation provided by the author while [14] utilized Julia and Flux ML. Training networks for the tabular datasets on $F_1$-Score using our method on a CPU took a median time of $2.3 \pm 0.16$ minutes whereas the adversarial approach on CPU took a median time of $70 \pm 118$ minutes.

## 5.2 Experiments on Image Data

We conducted experiments similar to the those in Sec. 5.1 with higher dimensional data using two different binary image datasets created from the CIFAR-10 dataset [22]. The CIFAR-10-Transportation and CIFAR-10-Frog datasets had a 40% and 10% positive class balance, respectively.

We focused on Accuracy and $F_1$-Score, given the limitations with AUROC found in earlier experiments. We also excluded the adversarial approach to optimize $F_1$-Score [14] due to long runtime on tabular data. Otherwise, we use the same set up from Sec. 5.1.

Overall, our findings with image data in Table 2 are consistent with tabular results from Sec. 5.1. Our method of optimizing $F_1$-Score over the soft-set confusion matrix performed better than traditional BCE for both datasets when evaluated on both $F_1$-Score and Accuracy. In Table 2, line (1) has higher scores than line (3). Results from our method of optimizing Accuracy over the soft-set confusion matrix (line (2) in Table 2) are better than the results for BCE (line (3) in Table 2).

## 5.3 Experiments on Graph Data

We applied our method to a more structured representation of the examples in the CocktailParty dataset [40]. Following Thompson et al. [35], we constructed a fully-connected graph in which each

Table 3: Graph data from the **CocktailParty** ($\mu \pm \sigma$) dataset classified using a Graph Neural Network (GNN) following Thompson et al. [35]. $F_1$ and Accuracy and via our proposed method. Bold indicates performance better than or equal to BCE baseline.

| | Loss | $F_1$-Score | Accuracy |
|---|---|---|---|
| (1) | $F_1$* | **0.83 ± 0.01** | **0.77 ± 0.01** |
| (2) | Accuracy* | 0.77 ± 0.02 | 0.72 ± 0.02 |
| (3) | BCE | 0.81 ± 0.03 | 0.75 ± 0.03 |

Table 4: **Mammography** ($\mu \pm \sigma$) dataset: $F_\beta$ ($\beta = \{1, 2, 3\}$) loss using the proposed method, to balance between precision and recall while maximizing $F_1$-Score.

| | Loss | $F_1$-Score | $F_2$-Score | $F_3$-Score | Precision | Recall |
|---|---|---|---|---|---|---|
| (1) | $F_1$* | 0.61 ± 0.06 | 0.57 ± 0.06 | 0.56 ± 0.06 | 0.70 ± 0.06 | 0.55 ± 0.07 |
| (2) | $F_2$* | 0.63 ± 0.04 | 0.67 ± 0.04 | 0.69 ± 0.04 | 0.57 ± 0.05 | 0.71 ± 0.04 |
| (3) | $F_3$* | 0.57 ± 0.03 | 0.69 ± 0.02 | 0.75 ± 0.02 | 0.44 ± 0.04 | 0.81 ± 0.03 |

person corresponded to a node in the graph and edges held distances between people. We then used a message-passing Graph Neural Network (GNN) [2] to predict pairwise affinities corresponding to whether two individuals were part of the same conversational group. This task was similar to the experiment on tabular data (Sec. 5.1) using the CocktailParty dataset, but instead of predicting interactions among pairs of potential interactants independently, we made predictions for all the pairs seen in a frame of the dataset simultaneously. Using the hyperparameters proposed by Thompson et al. [35], we obtained results (Table 3) in line with those for the tabular data (Sec. 5.1).

## 5.4 Balancing Between Precision and Recall

Our method also allows training-time optimization that balances between precision and recall using $F_\beta$-Score. This approach to training is particularly useful in real-world scenarios where there is a high cost associated with missed detections. This type of metric is difficult to optimize effectively for using a typical BCE loss because BCE is not aware of any preference towards precision or recall.

Results in Table 4 show that using the proposed method to optimize $F_\beta$-Score is an effective way of maintaining maximum classifier performance while balancing between precision and recall at a ratio appropriate for a given task. Increasing values of $\beta$ correspond to an increased preference toward recall with small loss of total performance measured by $F_1$-Score. Similar results on optimizing for $F_\beta$-Score with additional datasets are provided in Sec. 6.2 of the Supplementary Material.

## 6 Related work

Our work is inspired by research on the direct optimization of evaluation metrics for binary classification. This includes plug-in methods that empirically estimate a threshold for a classifier on a metric. For example, [26] demonstrated the applicability of plug-in classifiers to optimize $F_1$-Score with linear models. For metrics based on linear combinations of confusion matrix set cardinalities, [21] identified an optimal plug-in classifier with a metric-dependent threshold. Also, [20] explored the Precision@K metric for linear models in the context of ranking. Our approach is not a competitor to plug-in methods, but rather an approach to train a neural network classifier on a differentiable approximation of a metric based on the confusion matrix. As such, it could be used in conjunction with a plug-in method, if desired.

Works such as [7, 19] optimized specific metrics like Precision@K and $F_1$-Score in online learning, which is characterized by the sequential availability of data. Our work does not address online learning, but batch learning methods. Additionally, other work has focused on optimizing AUROC [18, 39], F-score [10, 41], and AUPRC in the context of ranking [12]. Rather than focusing on a single specific metric, we provide a flexible method for optimizing a neural network using approximations of varied metrics based on the confusion matrix. For example, our experiments in Sec. 5.4 show how tradeoffs between precision and recall can be made by adjusting the $\beta$ parameter of the $F_\beta$-score during training with our method.

In the field of computer vision, differentiable surrogate losses have been proposed for the $F_1$-Score [6, 8], Jaccard Index [3, 28, 29], and Dice score [4, 24, 27, 34]. However, since these methods are applied to difficult problems in computer vision, they incorporate a particular surrogate, such as a surrogate for the $F_1$-Score, into a larger composition of losses. Mean average precision (mAP) [13], for example, is composed of a number of intersection-over-union (IoU) values at different thresholds for multiple classes. We leave the study of metrics which require complex aggregation to future work. In our experiments, we focus on evaluating the impact of individual losses approximating a desired evaluation metric. We also evaluate the theoretical merits of optimizing a differentiable surrogate for confusion-matrix based metrics using an approximation of the Heaviside step function and soft-set confusion matrix values.

Recently, adversarial approaches have emerged as another related area of research. Wang et al. [37] used a structured support vector machine and reported performance on Precision@K as well as $F_1$-Score. Fathony and Kolter [14], which we compare against, improved performance via a marginalization technique that ensured polynomial time convergence. The authors evaluated their adversarial approach on Accuracy and $F_1$-Score, among other metrics, while reporting performance relative to BCE [14]. Downsides of the latter approach are that it is limited to a small batch size, on the order of 25 samples, and has cubic runtime complexity. Our approach does not limit batch sizes and has a worst-case runtime complexity equivalent to the runtime complexity of the confusion-matrix based metric being used as a loss.

## 7    Broader impact and ethics

Our work was motivated by the potentially wide ranging impact that more flexible and robust binary classifiers could have across application domains such as social group dynamics, sociology, and economics, which we explored in our experiments. Improving the tools used in these areas of research has the potential to positively impact human quality of life, but are still susceptible to data bias. These tools should be used with care when building safe and responsible artificial intelligence systems.

## 8    Conclusion

We proposed a novel method to optimize for confusion-matrix based metrics by using a Heaviside function approximation and soft-set membership. Our method addresses the common training-evaluation gap when working with binary neural network classifiers: these networks are typically trained with BCE loss, but evaluated using a confusion-matrix based metric, such as $F_1$-Score.

Our theoretical analyses showed that soft-set confusion matrix based metrics, such as $F_1^s$, are Lipschitz continuous and are likely to converge in expectation to the true metric's expectation. Additionally, our experiments showed the feasibility of using our method to optimize confusion-matrix based metrics. While many factors play into final classifier performance, such as balance of samples in the dataset, the desired evaluation metric, the approximation of H, and network hyperparameters, we found that in many cases, final classifier performance can be improved by bridging the training-evaluation gap using our method. In fact, optimizing model parameters for $F_1$ with soft sets resulted in better evaluation results with $F_1$-Score and better or comparable evaluation results with Accuracy and AUROC. Our approach also outperformed other methods that directly optimize for a specific metric. For example, Fathony and Kolter's [14] adversarial approach applied to $F_1$-Score had much longer training time and resulted in worse performance than our approach.

In the future, we are interested in further investigation of other aspects of our approach such as analyzing statistical sample efficiency as well as the approximation error and the generalization error. Further, we believe studying how our method can be applied to multi-class classification problems is an interesting area for future work.

## 9    Acknowledgements

The authors thank Andre Wibisono for helpful discussions and feedback. This work was supported by the National Science Foundation (NSF), Grant No. (IIS-1924802). The findings and conclusions in this article are those of the authors and do not necessarily reflect the views of the NSF.

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
