# OpenReview forum: "Bridging the Gap: Unifying the Training and Evaluation of Neural Network Binary Classifiers"
_NeurIPS.cc/2022/Conference — NeurIPS 2022 Accept_

### Official Review · Reviewer_x14X · 2022-07-10

**Rating:** 7
**Confidence:** 4
**Soundness:** 3 good
**Presentation:** 3 good
**Contribution:** 3 good

**Summary:**

This work focuses on the loss function of the binary classification problem and proposes the use of the differential Heaviside Function Approximation $\mathcal{H}$ and soft set membership as a loss to directly optimize desired confusion matrix-based metrics such as accuracy, F1-score, etc., instead of the commonly used surrogate objective BCE. The study focuses on the analysis of a sigmoid function and a five-point linearly interpolated function as an approximation, with some assumptions providing theoretical underpinning. Experiments are conducted on several unbalanced level datasets, and the results are consistently good or comparable in some scenarios.

**Questions:**

Results are observed when the evaluation measure is F1 and the loss is also F1*, however practically all studies on accuracy are comparable to BCE (tabular datasets) or the same improvement is reached using F1* and Accuracy* (image datasets). Also, it appears that F1* performs better on numerous evaluation metrics; does this contradict what the theory predicts?

**Limitations:**

Did not conduct research with human subjects

**Strengths And Weaknesses:**

Strengths: The strategy suggested in this article provides a number of advantages over earlier attempts to directly improve measures. It alleviates the batch size limitation and being faster for the adversarial approach [11] (cubic runtime complexity), and it supports a wider variety of metrics as opposed to WMW [33].

Specifically, the method can optimize metrics such as $F_1$, $F_\beta$, Accuracy, AUROC, etc. directly and balance the tradeoff of Precision and Recall, without increasing the training time complexity (linear).

The authors provide a theoretical foundation for this method and empirically validate it using a variety of datasets with varying levels of imbalance. The linear approximation function was introduced in [27] as an activation function, but this work seems the first to use it in the training objective.


Weaknesses: The experimental results show that only some of the metrics have a significant improvement, while others do not. For example, AUROC has no unbiased estimator which leads to bad performance. This approach seems less directly applicable to multiclassification tasks.

---

> ### Author Response · Authors · 2022-08-02
> **Response to Reviewer x14X**
>
> Thank you for taking the time to provide a thoughtful review. We are also excited that the proposed method provides numerous advantages over existing methods. Our answers to the questions in the review are provided below along with our proposed changes to incorporate the feedback into the paper.
>
> __Do the results about F1* and Accuracy* contradict what the theory predicts? (Question 1, Weakness Sentence 1)__
>
> We believe the experimental results do not contradict the theoretical results. Our theoretical analysis says that our proposed loss should converge to the evaluation metric, but nothing prevents binary cross entropy (BCE) to result in comparable performance in some cases. In general, though, our theoretical analyses do not lead us to believe that BCE would converge to any desired metric based on confusion matrix values in the limit. In particular, binary cross entropy loss is expressed as:
> $$-\\frac{1}{n}\\sum_{i=1}^n \\left(y_i \\log p + (1-y_i)\\log (1-p) \\right)$$
>
> while Accuracy is:
> $$\\frac{|TP| + |TN|}{n} = \\frac{\\sum_{i=1}^n \\left (y_i \\hat y_i^{\\mathcal{H}} + (1-y_i)(1-\\hat y_i^{\\mathcal{H}})\\right)}{n}$$
> These are different expressions that don’t generally converge to each other as $n \to \infty$. We will clarify this important point in our theoretical analysis (Section 4).
>
> It is also worth noting that the Accuracy metric is particularly challenging, as it may overestimate classifier performance for imbalanced datasets [14] (lines 208-209). We believe that for this reason, $F_1$-Score is often a better indicator of a classifier’s true performance and, therefore, optimizing $F_1$-Score using our method can result in better performance than optimizing for other confusion-matrix based metrics with our approach. This highlights an interesting challenge in the evaluation and training of neural networks: how does one identify the best evaluation metric and thereby choose a suitable training loss using our method? This is an important question that warrants further research in the future.
>
>
> **Is this approach less directly applicable to multi-class classification? (Weakness Sentence 3)**
>
> We agree that the problem of multi-class classification presents an interesting opportunity for future work. As noted in our response to Reviewer 7RmE under the heading titled “Multiclass classification”:
>
> There is promise in applying our approach to multi-class classification, however this requires choosing a metric to apply our method to. In multi-class classification, common metrics (such as F1) can be computed using an aggregation strategy like macro or micro averaging. In these cases of simple differentiable aggregations, it is straightforward to apply our method by approximating the Heaviside function, computing soft sets, and then computing the final score. In other cases, further research is required for more complex aggregations. For example, Top-N accuracy considers a response correct if it is included in one of the top N predictions. In computer vision, another complex example is the mean average precision at a number of intersection-over-union (IoU) thresholds (as proposed by Everingham et al.). We will briefly discuss these ideas in the related work section of our paper to inspire future extensions of our work.
>
> Everingham et al. (2015). The pascal visual object classes challenge: A retrospective. IJCV.

---

> > ### Comment · Reviewer_x14X · 2022-08-08
> > **Response to rebuttals**
> >
> > Thank you for your informative responses, I have read it, along with other reviewers' opinions, throughs.  I would like to maintain my positive ratings.
> >
> > I would also like to hear the opinion from reviewer MHi1.

---

### Official Review · Reviewer_MHi1 · 2022-07-10

**Rating:** 4
**Confidence:** 2
**Soundness:** 3 good
**Presentation:** 2 fair
**Contribution:** 2 fair

**Summary:**

In binary NN classifiers, we train the models on BCE objective but at the test time, we evaluate the model's accuracy or AUROC instead. The authors propose a differentiable approximation to the test time metrics (like Accuracy/AUROC/F1 score) so that we can use them for training the model as well. The authors evaluate the new loss function on 5 datasets with varying levels of class imbalance and show that the models trained with proposed loss have better generalization.

**Questions:**

The following are my major concerns, it would be great to discuss this
-	How expensive is the method? In terms of compute and time efficiency.
-	How robust is the model to slight OOD perturbations?
-	Is this method scalable to large datasets?


**Strengths And Weaknesses:**

Strengths:
- The introduction of a new optimization objective.
- The investigations seem thorough.

Weaknesses:
- Did not justify enough why this new optimization objective is useful. I understand this improves the metrics we are interested in measuring but are there any other benefits.
- The paper is hard to follow in some places.

---

> ### Author Response · Authors · 2022-08-02
> **Response to Reviewer MHi1**
>
> Thank you for your review – we are glad you found our investigation thorough. Below we address the weaknesses of the paper that are pointed out in the review and answer the questions that are major concerns.
>
> **Why is the proposed optimization objective useful? (Weakness 1)**
>
> Our new optimization objective is important for three reasons. First, in many cases, classifier performance is improved by bridging the training-evaluation gap using our method, as you noted. Second, as discussed in Section 6, our method is different from much prior work in that our approach is flexible and can be used with a variety of metrics composed of  confusion matrix values. Third, our approach makes improvements with respect to runtime of related work and our approach does not limit batch size which facilitates shorter training times (Section 6). These improvements make our approach more practical than existing alternatives.
>
> **Clarifications (Weakness 2)**
>
> In response to your comment that the paper was hard to follow in some places (Weakness #2), we address your specific questions below. We’d be happy to discuss any other areas where the paper was hard to follow in the author-reviewer discussion period.
>
> **How computationally expensive is the method? (Question 1)**
>
> Our approach has the worst-case runtime complexity of the confusion-matrix based metric being approximated as a loss (as mentioned briefly on lines 315-316 of the paper). Computation of such metrics are linear with regard to the number of samples, which is discussed in Section 3 of the Supplementary Material. In contrast, the adversarial method by Fathony and Kolter has cubic runtime complexity [11].
>
> **How robust is the model to slight OOD perturbations? (Question 2)**
>
> Our work makes no claims regarding out of distribution data. However, our theoretical results show asymptotic equivalence (for large $n$) between the proposed soft-set confusion-matrix based loss (such as $F_1$*) and the true metric ($F_1$-Score).
>
> As noted in our response to Reviewer 7RmE under the heading titled “Approximation error and generalization error”:
>
> Analyzing the approximation error and generalization error is an interesting direction for future work. However, please note that such analyses typically consider the loss used to train the classifier — which is the focus of our paper — along with the neural network architecture (Cao & Gu, 2019) and even properties of the data (Nakada & Imaizumi, 2020). We will mention analyzing these errors as valuable future research directions in the conclusion section.
>
> Cao & Gu (2019). Generalization bounds of stochastic gradient descent for wide and deep neural networks. NeurIPS.
>
> Nakada & Imaizumi (2020). Adaptive Approximation and Generalization of Deep Neural Network with Intrinsic Dimensionality. J. Mach. Learn. Res.
>
>
> **Is this method scalable to large datasets? (Question 3)**
>
> Yes, our proposed method is scalable to large datasets. The typical runtime complexity of our approach is typically linear wrt. number of samples, as mentioned earlier in this response. Also, in contrast to the work by Fathony and Kolter [11], we place no constraints on batch size during training, as noted on line 313 of our paper. This means that one can train with as big of a batch size as GPU memory allows, potentially shortening training time.

---

### Official Review · Reviewer_Mx4J · 2022-07-12

**Rating:** 6
**Confidence:** 3
**Soundness:** 3 good
**Presentation:** 4 excellent
**Contribution:** 3 good

**Summary:**

The paper proposes a method to directly optimize evaluating metrics, such as F1 and AUROC, at training time. The method includes a piecewise linear approximation of the 0-1 loss (heaviside function), and soft versions of F1 and AUROC metrics based on the diffierentiable approximation. Experiments show that the proposes method can achieve superior F1 and AUROC compared to models trained with binary cross entropy loss.

**Questions:**

1. Based on my understanding, I think the technical part of proposed method includes a piecewise linear approximation of the heaviside function, and approximations of confusion-matrix metrics based on soft sets. I'd like to see an ablation study where these two metrics are evaluated separately. For example, if the heaviside function is approximated with sigmoid function, does the proposed "accuracy" metric coincides with BCE?

2. Sec. 4.2 presents a consistency result. Isn't that also true for the BCE loss, at least for accuracy? I think to firmly demonstrate the advantage of the proposed method, consistency alone might not be enough, and one might want to consider the statistical (sample) efficiency?

**Limitations:**

Limitations an societal impact are sufficiently addressed.

**Strengths And Weaknesses:**

Strength
- Binary classification is an important and frequent problem.
- The proposed method can have wide application potential.
- The proposed method is simple and effective.

Weaknesses
- The novelty might be somewhat thin: approximating non-diffierentiable functions with differentiable ones are common. The theoretical grounding section might not be too superising (continuity, consistency). Proposing to directly optimize evaluating metrics at training time might be novel, but I think it is also studied, as those mentioned in related works and are compared against.

---

> ### Author Response · Authors · 2022-08-02
> **Response to Reviewer Mx4J**
>
> Thank you for your detailed review. We agree that this is an impactful area of research and we are excited about the simplicity and effectiveness of our approach. Below we address the questions about our approach and describe the changes that we will make to the paper to improve its clarity and clarify our contributions.
>
> **What is the novelty of the proposed approach? (Weakness 1)**
>
> Our work is novel in that:
> 1. To the best of our knowledge, we are the first to combine a differentiable approximation of the Heaviside function with a probabilistic view of the typical confusion-matrix values using soft sets in order to train binary neural network classifiers using losses that approximate typical classification metrics composed of confusion matrix values.
> 2. Our approach is not only simple to implement, but it is flexible in that it can be applied in a straightforward manner to any classification metric composed of confusion-matrix values.
> 3. Our approach makes improvements in runtime performance with respect to related work and without limiting batch sizes in practice (Section 6).
>
> **Does the sigmoid approximation of the Heaviside function change performance of the approach? Could you conduct an ablation study for the two components of the proposed approach? (Question 1)**
>
> Our proposed approach uses a differentiable approximation of the Heaviside step function to build a probabilistic confusion matrix using soft sets. We can use a piecewise linear approximation for the approximation of the Heaviside function or  a sigmoid approximation (see results in Section 2 and Section 5 of the Supplementary Material). The important idea here is to approximate the Heaviside function in a way that satisfies the properties explained in Section 3.1 of the paper. Unfortunately, it is not possible to ablate (to remove) the use of either soft sets or a differentiable approximation of the heaviside function in our implementation because both are required for a differentiable objective.
>
> **Should consistency of BCE be considered? Should we consider statistical (sample) efficiency? (Question 2)**
>
> In Section 4.2, we consider a classifier trained under our setup optimizing over $F_1^s$, which is the $F_1$-Score computed with soft sets. We prove that $F_1^s$ and $F_1$, which is the true $F_1$-Score, both converge almost surely to the same expression as $n \to \infty$. This similarly holds for other metrics such as Accuracy and AUROC. However, if a network is optimized using binary cross entropy loss and then evaluated using a confusion-matrix based metric, the convergence result no longer holds. For example, binary cross entropy loss is expressed as:
> $$-\\frac{1}{n}\\sum_{i=1}^n \\left(y_i \\log p + (1-y_i)\\log (1-p) \\right)$$
>
> while Accuracy is:
> $$\\frac{|TP| + |TN|}{n} = \\frac{\\sum_{i=1}^n \\left (y_i \\hat y_i^{\\mathcal{H}} + (1-y_i)(1-\\hat y_i^{\\mathcal{H}})\\right)}{n}$$
>
> While training using binary cross entropy loss generally improves evaluation accuracy, these are different expressions that don’t generally converge to each other even as $n \to \infty$.
>
> While our experimental results show promising results, we agree that future work could investigate more theoretical aspects of our approach, such as statistical sample efficiency. We will briefly mention this idea in the conclusion section of the paper as another interesting future research direction.

---

### Official Review · Reviewer_7RmE · 2022-07-19

**Rating:** 6
**Confidence:** 4
**Soundness:** 3 good
**Presentation:** 3 good
**Contribution:** 3 good

**Summary:**

This paper proposes to use a differentiable approximation of the Heaviside step function (which determines where the prediction should be positive according to the input and threshold) to build a loss function that theoretically approximates Accuracy or F-score.  The motivation is to bridge the gap between training loss and evaluation metrics in binary classification.

In terms of the method, it has two parts. The first is to approximate the Heaviside function with a sigmoid or piecewise linear function with hyperparameter tau. Then this paper introduces the soft set membership to measure the degree of belonging to a certain confusion matrix set. In terms of theoretical support, the author proves that the new approximation of the Heaviside function is Lipschitz continuous and the loss constructed by their combination also makes the variation of each step in SGD optimization small. Then the paper proves that the loss that approximates the F1 metric converges to the F1 metric when the number of samples is infinite. Finally, in the experimental part, the paper experiments on tabular and image data and also explores the balance of precision and recall. The approximated loss of F1 score achieves higher performance than BCE loss and other losses.


**Questions:**

1. I suggest that this paper can give a figure that pairs and compare the original version and soft version and highlight the difference. Formula 2 and Formula 6 look identical (the difference is the small s).

2. Line 124, “By [11]”, I recommend not making numbers the subject of a quote.

3. What are the challenges of extending this research to multiclass classification?

4. I suggest that this article can also experiment on more kinds of data, such as binary classification on graph data.


**Ethics Review Area:**

["I don’t know"]

**Limitations:**

The paper discusses limitations scattered throughout the paper. Besides the limitation summarized in checklist 1 (d), they also acknowledged other limitations.

In Line 119, the paper acknowledges that the mini-batch stochastic gradient descent (SGD) optimization method does not provide an unbiased estimator on non-decomposable metrics.

In Line 202, the paper acknowledges that the F1-Score computed from soft sets is a biased estimator for the expected true F1-Score for finite n.

In Line 249-254, they find that the loss that approximates AUROC does not work.


**Strengths And Weaknesses:**


Strengths:
1. The paper is novel and compares some related methods in related work and experiments.
2. This is a completed work, with both theoretical support and experimental results, and the authors honestly point out various deficiencies.
3. The presentation of this article is very good; the overall structure and specific details are relatively clear.
4. This paper provides a training loss that approximates the evaluation metric. The difference between training and evaluation is a problem of great interest.

Weakness：

1. Although this paper has compared adversarial-based methods in experiments, comparisons with other differentiable surrogate losses are lacking.

2. This paper can compare previous methods and the method in this paper for the difference in approximation error.  Previous methods include BCE loss and the previous methods to solve the difference between training loss and test indicators.

3. What kind of theoretical problems will the gap between the two-class training loss and test indicators bring about; this paper lacks the analysis in this regard.

4. There are some differences between the test sample and the training distribution. The closeness of the loss at training time to the metric at test time does not seem to solve this problem, and this article does not consider generalization error.

---

> ### Author Response · Authors · 2022-08-02
> **Response to Reviewer 7RmE**
>
> Thank you for the thoughtful review.
>
> **Comparisons with other differentiable surrogate losses (Weakness 1)**
>
> To the best of our knowledge, our approach is the only published method with the flexibility to train neural network binary classifiers using losses that approximate *any* classification metric composed of confusion-matrix values. Therefore, we compared against specific surrogate approaches for well-established classification metrics, including the adversarial approach for F1 proposed by Fathony and Kolter [11] and the WMW statistic [33] (a differentiable method for optimizing the AUROC score). Additionally, Sec. 7 of the Supp. Material includes results against other baselines, including the differentiable surrogate for DICE ([8] in the Supp. Material).
>
> **Approximation error and generalization error (Weaknesses 2 & 4)**
>
> Analyzing the approximation error and generalization error is an interesting direction for future work. However, please note that such analyses typically consider the loss used to train the classifier — which is the focus of our paper — along with the neural network architecture (Cao & Gu, 2019) and even properties of the data (Nakada & Imaizumi, 2020). We will mention analyzing these errors as valuable future research directions in the conclusion section.
>
> Cao & Gu (2019). Generalization bounds of stochastic gradient descent for wide and deep neural networks. NeurIPS.
>
> Nakada & Imaizumi (2020). Adaptive Approximation and Generalization of Deep Neural Network with Intrinsic Dimensionality. J. Mach. Learn. Res.
>
> **Gap between the training loss and test indicators (Weakness 3)**
>
> Empirical results (e.g., [10]) show that the training-evaluation gap hinders evaluation performance. The gap means that in many applications, the metric differs significantly from the surrogate loss [30] and there may not be a strong correlation between minimizing a surrogate loss and improving an evaluation metric [26].  Also, it could mean that classifiers are solving “the wrong problem” when optimizing a surrogate loss, leading to suboptimal evaluation performance [15]. We will clarify this point in the introduction of the paper.
>
> **Multiclass classification (Question 3)**
>
> There is promise in applying our approach to multi-class classification, however this requires choosing a metric to apply our method to. In multi-class classification, common metrics (such as F1) can be computed using an aggregation strategy like macro or micro averaging. In these cases of simple differentiable aggregations, it is straightforward to apply our method by approximating the Heaviside function, computing soft sets, and then computing the final score. In other cases, further research is required for more complex aggregations. For example, Top-N accuracy considers a response correct if it is included in one of the top N predictions. In computer vision, another complex example is the mean average precision at a number of intersection-over-union (IoU) thresholds (as proposed by Everingham et al.). We will briefly discuss these ideas in the related work section of our paper to inspire future extensions of our work.
>
> Everingham et al. (2015). The pascal visual object classes challenge: A retrospective. IJCV.
>
> **Experiments on graph data (Question 4)**
>
> We performed an additional experiment that will be added to our Supp. Material Sec. 6. Following Thompson et al. and using the CocktailParty dataset ([14] in the Supp. Material), we constructed a fully-connected graph in which each person corresponded to a node in the graph. Edges corresponded to metric distance between people. We then used a Graph Neural Network to predict ​​pairwise affinities that encoded whether or not two individuals were part of the same conversational group. This task was effectively the same as described in Section 4.1.2 of our Supp. Material, but instead of predicting interactions among pairs of potential interactants independently, we made predictions for all the pairs seen in a frame of the dataset simultaneously. The results of this additional experiment, which used the hyperparameters proposed in Thompson et al., are in line with the existing results reported in our paper:
>
> ||Loss|$F_1$-Score|Accuracy|
> |---|---|---|---|
> |(1)|$F_1$*|$0.83\pm0.01$|$0.77\pm0.01$|
> |(2)|Accuracy*|$0.77\pm0.02$|$0.72\pm0.02$|
> |(3)|BCE|$0.81\pm0.03$|$0.75\pm0.03$|
>
> Thompson et al. (2021). Conversational group detection with graph neural networks. ICMI.
>
> **Other improvements**
>
> Due to space constraints, we will highlight the difference between the confusion matrix and soft confusion matrix formulation in the text (**Question 1**). Immediately after Eq. 6, we will add: “Whereas the values of the confusion matrix (tp, fp, tn, fn), are the sum of zeros and ones, the values of the soft confusion matrix (tp_s, fp_s, tn_s, fn_s) are the sum of continuous values in $[0,1]$ (rather than integers).”
>
> We will avoid making numbers the subject of a quote (**Question 2**). Thanks.

---

> > ### Comment · Reviewer_7RmE · 2022-08-09
> > **Post-rebuttal Response by 7RmE**
> >
> > Thank you for your detailed response. I also read your response to other reviewers. I decide to maintain my positive rating.  I hope this paper could include an analysis of generalization errors, studies on multiclass classification, and a comparison with other surrogate losses in the main content of the paper.

---

### Meta-Review · Area_Chair_aUvv · 2022-08-27

**Recommendation:** Accept
**Confidence:** Certain

**Metareview:**

When training binary classifiers, one usually minimizes the (surrogate) binary cross-entropy loss (BCE), but evaluates on metrics such as F1, AUROC, or other confusion metric-based scores. The authors propose to instead combine a differentiable approximation of the Heaviside function with a probabilistic view of the typical confusion matrix values using soft sets to directly optimize F1 and AUROC at training time. The authors show that under certain assumptions the metrics computed over the soft-set confusion matrix values are asymptotically similar to the underlying true metric. Finally, the authors evaluate the proposed approximations on several unbalanced datasets and show competitive performance with respect to optimizing BCE.

While the reviewers outlined some weaknesses, they agreed that this work is relevant to the larger research community and presents a novel approach with potential to be used in practice. During the rebuttal phase the authors addressed the main remaining issues and I will recommend acceptance. Please incorporate all the information presented during the rebuttal phase into the manuscript.

**Award:**

No

---

### Decision · Program_Chairs · 2022-09-14

Accept